# Modeling a Microtubule Filaments Mesh Structure from Confocal Microscopy Imaging

**DOI:** 10.3390/mi11090844

**Published:** 2020-09-10

**Authors:** Yutaka Ueno, Kento Matsuda, Kaoru Katoh, Akinori Kuzuya, Akira Kakugo, Akihiko Konagaya

**Affiliations:** 1National Institute of Advanced Industrial Science and Technology, 2-4-7 Aomi, Koto-ku, Tokyo 135-0064, Japan; k-katoh@aist.go.jp; 2Graduate School of Chemical Sciences and Engineering, Hokkaido University, North 10 West 8, Kita-ku, Sapporo 060-0810, Japan; mattuan1112@eis.hokudai.ac.jp (K.M.); kakugo@sci.hokudai.ac.jp (A.K.); 3Department of Chemistry and Materials Engineering, Kansai University, 3-3-35 Yamate-cho, Suita, Osaka 564-8680, Japan; kuzuya@kansai-u.ac.jp; 4School of Computing, Department of Computer Science, Tokyo Institute of Technology, 4259 Nagatsuta-cho, Midori-ku, Yokohama 226-8502, Japan; konagaya@molecular-robotics.co.jp; 5Molecular Robotics Research Institute, Ltd., Kyowa Create Dai-ichi build. 3F, 3-11-1 Shibaura, Minato-ku, Tokyo 108-0023, Japan

**Keywords:** artificial muscle, microtubule, kinesin, fluorescent microscopy, molecular robotics, molecular machine

## Abstract

This study introduces a modeling method for a supermolecular structure of microtubules for the development of a force generation material using motor proteins. 3D imaging by confocal laser scanning microscopy (CLSM) was used to obtain 3D volume density data. The density data were then interpreted by a set of cylinders with the general-purpose 3D modeling software Blender, and a 3D network structure of microtubules was constructed. Although motor proteins were not visualized experimentally, they were introduced into the model to simulate pulling of the microtubules toward each other to yield shrinking of the network, resulting in contraction of the artificial muscle. From the successful force generation simulation of the obtained model structure of artificial muscle, the modeling method introduced here could be useful in various studies for potential improvements of this contractile molecular system.

## 1. Introduction

In nanotechnology, the mobile micromachine is gaining attention for potential application to the development of artificial muscle [1,2,3]. In particular, materials with microtubules and motor proteins have been demonstrated to provide contraction [4]. Motor proteins were designed to produce the pulling force between microtubule filaments that is known to happen in cells. The molecules for the artificial muscle were designed to organize contractile systems as an intrinsic characteristic, so that a muscular structure is built by self-organization. This material was made possible by using DNA origami technology [5,6] to bind microtubule filaments to form the aster structure. In order to improve contraction strength and the control of force generation, improvements based on observation of its structure are needed.

In this study, we employed confocal laser scanning microscopy (CLSM) with a high-resolution imaging method [7] to observe the 3D structures of the microtubule network of the reported artificial smooth muscle. The high-resolution images of the microtubules illustrated their soft structural characteristics and contraction ability. The 3D volume data was fine enough to annotate the cylinders of microtubules. We then built a mesoscale structural model of this microtubule network and attempted to simulate the contractile motion of the microtubule filaments pulled by motor proteins.

The modeling study requires the proper computational tool for observation of 3D volume data and manipulation of the polygon mesh models in 3D scene graphics on the nanoscale range. An atomic model of the structure of kinesin, a motor protein that moves along microtubule filaments, was available at the Protein Data Bank, and is represented in volume shape data as a polygon mesh. For modeling tasks such as layout and molecule orientation, we introduced Blender [8], a general-purpose 3D graphics modeling software. We describe here how the modeling and functions of this software tool were effective in building our model. Having a structural model of the microtubule network used in artificial muscle enables understanding of how the motor proteins pull the microtubule filaments, and allows demonstration of the movement using the rigid body physics simulation function in Blender. Although a realistic simulation of the contraction event of the material will require more investigation of the motor protein behavior, we were able to move the microtubule filaments towards each other in a virtual scene graphics environment. The model could be used to improve the study of artificial muscle and to guide the design of effective force generation and control of contraction events.

This paper reports our recent high resolution imaging results for the artificial muscle, using CLSM. We then begin the modeling study of the microtubule mesh structure using Blender to evaluate the characteristics of the dynamics and we explore possible ways to improve the target molecular system. Currently, we well reproduced contraction of this artificial muscle that is initiated upon ATP addition, as described in the previous report [4]. We will also discuss a potential strategy to accomplish multiple time contractions by a quick recovery method based on the obtained dynamic property of this contractile material.

## 2. Materials and Methods

### 2.1. Preparation of Artificial Muscle

The artificial muscle specimen, composed of hierarchically ordered microtubule asters, was prepared as reported previously [4]. Short microtubule seeds were polymerized from tubulin dimers labelled with fluorescent dye Alexa 488. The seed microtubules were elongated through polymerization of azide-conjugated tubulins and tubulin dimers that were labelled with TAMRA dye. Single strand DNA with a predesigned sequence was conjugated to the elongated part of the microtubules through a click reaction. According to the standard procedure as reported in literature [9] rhodamine or Alexa488-labeled tubulin was prepared using 5/6-carboxytetramethylrhodamine succinimidyl ester (TAMRA-SE; Invitrogen, Carlsbad, CA, USA) or Alexa488, respectively, with the standard technique, and the labeling ratio was 1.0 in each case. Azide-labeled tubulin was prepared using N3-PEG4-NHS, following the protocol of labeling tubulin with fluorescent dye. After preparation, the DNA-conjugated microtubules were mixed with DNA origami to form asters. Upon addition of kinesin multimer linkers created with streptavidin molecules binding four kinesin dimers, and with the addition of ATP, the filament mesh structure contracted like muscle. We further examined this contraction scheme in our model calculation described below.

### 2.2. Imaging with Confocal Microscopy

A CLSM system (A1 Nikon) with an oil-immersion objective lens (Plan Apo λs 60× N.A. 1.4) and two excitation laser lines (487.2 and 561.5 nm) was configured for the setting suitable for 3D deconvolution (a narrow pinhole radius 0.3 Airy unit) [7,10]. The obtained series of z-stack images, with 60 nm pixels and a 100 nm depth interval, was then processed by the deconvolution feature in the NiS Elements-AR software, using the 3D-auto setting developed by Nikon Inc. (Tokyo, Japan). After the deconvolution, image resolution was improved better than the 180 nm discussed in the literature [11], which allowed observation of the filament structures of microtubules [7]. The Richardson-Lucy algorithm was also tested, and produced largely similar results, which were applied to the images in Figure 1.

The antifade mountant ProLong Diamond (ThermoFisher, Waltham, MA, USA), diluted 10% with 80 mM PIPES buffer, was also added, as in the microtubule specimen.

### 2.3. Building the Microtuble Filament Model from Volume Data

The volume data for the two separated channels were loaded into Image J (version 1.51n, Fiji) and the isosurface polygon mesh data were calculated. Voxels were scaled to 8-bit density values, then displayed in the 3D View window at appropriate values through visual inspection. The isosurface polygon mesh data were saved in STL format, and the files were imported into Blender. We used Blender (version 2.79, Blender foundation, Amsterdam, The Netherlands) [8] for modeling work using primitive cylinders for microtubules. The 3D Blender workspace requires understanding of the functionality of Blender; we also prepared useful scripts for modeling macromolecules, which are available on our website [12] and various scientific studies [13,14]. All analysis could be conducted on a computer with a core i7-3770s CPU, 32 GB memory and an NVIDIA GPU 1050.

### 2.4. Rigid Body Dynamics Calculation

A rigid body physics calculation function is available for geometrical model building in Blender; it embeds the code of the physics engine Bullet [15]. The behavior of general mesh objects can be simplified with a cube or sphere, rather than strictly calculating a precise rigid body of mesh geometry. The integration of Newton’s equation is performed for the assigned geometrical objects (the number of objects should be no more than 1000, empirically). Using a cylinder model of microtubule filaments, kinesin multimer models were introduced to bind the filaments.

Dynamics calculation functions are insufficient for a realistic simulation of the molecular interaction between microtubules and kinesin using binding and releasing. We selected a motor primitive that moves one object to another at constant velocity.

## 3. Results

The image shown in Figure 1 is consistent with one from a previous report [4], with much improved contrast and resolution. Microtubule filaments intersect each other at slant angles, so there appears to be substantial room between filaments. The images look like soft mesh objects that would be able to shrink in volume.

Determining that the primary model of this artificial muscle was an aster-like assembly of the microtubules mediated by the binding DNA origami molecules was not easy. The observed structure appeared to be fragments of asters stacked together. We therefore also took images of a dilute concentration of microtubules at 0.1 μM, as shown in Figure 1b. These looked adequate to model the marginal gathering mesh of the microtubules, although the specimen did not work well for the contractile function. There were fragments of the aster structure densely packed and overlapped slantwise without an ordered pattern. Since individual filaments were visualized well, we obtained 3D volumes of selected network arrangements of microtubule filaments and built filament models of microtubules that fit the observed volume. Without the kinesin linkers, the mesh structure images were also used to reproduce the image in our previous report [4] as in Figure 1c, however it was difficult to obtain stable images due to fluctuations.

We consider that the observed network structure of microtubules maintains the same filament connection as the aster structure that we designed for the artificial muscle. Figure 2 illustrates the two aster structures that interact with the multimeric kinesin linker, allowing pulling of the two asters toward each other to produce contractile ability. While the complete aster structure could not be clearly observed, we found fragments of the aster structure in our imaging due to the dissociation of some filaments.

### 3.1. 3D Imaging of Microtubule Network

Figure 3 shows a typical structure of the microtubule network. 3D volume data were recorded by resonant scan function provided by the Nikon Ti A1. The depth slice image stacks were deconvolved and yielded reasonable high-contrast filament shapes that were in good accord with volume rendering observed by the NiS element viewer. The red regions stained by fluorescent dye TAMRA show connecting DNA origami molecules between microtubules. The network structure was also cross-linked by binding kinesin multimers in the green region of the image, where those kinesins were not visible.

Frequently the structure of the microtubule network fluctuated by Brownian motion, and sometimes drifted in the altering depth slice section recording. A quick scanning setting for image accumulation was necessary, but with increased image data noise. This could possibly be reduced by the addition of antifade mountant to fasten the specimen.

### 3.2. Fitting Microtubules into the 3D Volume

With the extracted isosurface mesh data, the obtained volume data was interpreted as a cylinder model of microtubule filaments. Figure 4 shows results that nicely fit the observation. Most of the volume was represented by straight cylinders, 0.5 μm in diameter and 4-μm long, and longer filaments of 7.5 μm. Some of the filaments were connected to each other and were regarded as a part of the aster structure. In order to fit the cylinder to the 3D volume data, cylindrical features several pixels in width were preferred; otherwise the volume connection would be discontinuous. Higher resolution volume data could obtain cylinder objects at much smaller radii; however, such fine voxels increased noise. We finally took the voxel data at 120 nm intervals for the isosurface volume data.

DNA origami binding regions are marked in red in Figure 4b, using the volume observed by another fluorescence channel in CLSM by visual comparison. It was difficult to build a solid model for DNA origami linker regions because sometimes the inside volume did not follow a set of rigid cylinders and may have included small chunks that perhaps indicated aggregates of the smaller molecules. The detailed fine structure could not be recovered precisely by the deconvolution process due to the limitations of the optical imaging resolution. Further improvements in modeling the fine structure could be achieved by introducing a higher resolution imaging method.

### 3.3. Dynamics of Microtubule Filaments

In order to evaluate the motility characteristics of the observed network structure of microtubule filaments, the simple motion of the model was evaluated by a rigid body motion calculation available in Blender. Although the introduced kinesin multimer has four arms, we first calculate two arms of the kinesin heads, pulling two microtubule filaments.

The basic motility element of the multimeric kinesin linker that attracts the two microtubule filaments toward each other in Figure 2 was constructed as shown in Figure 5a. Two filaments were positioned close enough to be linked by the kinesin multimer. The kinesin molecule moves toward the minus end of the microtubule filaments that is placed on opposite sides of this element. With this motility element, the kinesin motion toward the minus end of the microtubule filaments will move two filaments closer (Figure 5b,c).

The motion of the kinesin multimer was applied to slide one microtubule filament using a motor primitive in the physics engine module in Blender. Although single filament motion was a simple straightforward physics motion in Blender, having the two filaments interact was not easy. Introducing a virtual node that connected two kinesin motions, two motions of the kinesin head molecules were applied to two different microtubule filaments. The result was not satisfactory because the two microtubule filaments moved away and separated. We therefore introduced another “piston” constraint for the kinesin head to always attach onto the microtubule filament. Then the motion was satisfactory, pulling the two microtubule filaments toward each other. In this calculation, the object size was chosen to fit the default environment of Blender in arbitrary units: a cylinder length of 4.0 with diameter 0.2, a spherical kinesin head diameter of 0.1, and a motor speed 0.3 that starts at the time frame. The velocity damping factor 0.4 was applied for the kinesin head and a factor of 0.05 was applied for the microtubule cylinders; weights are 1.0 for both microtubule and kinesin heads. The calculation time step was 1/24 s and 200 frames resulted in behavior calculations for ~8 s. The model was tuned experimentally to yield natural behaviors of its objects, although it was a virtual world that does not correspond to the real object scale. The result of this configuration implemented in a Blender script file is available (at https://github.com/uenoyt/abam/ [16], including movies and construction details).

A partial fragment of the aster structure of the artificial muscle was also modeled by connecting microtubule filaments at both ends of this motility element. As shown in Figure 5d–f, the pulling motion of the motility elements working in the network structure successfully shrank the mesh structure to yield contraction of the artificial muscle. The motion of this network structure model is available as a movie file [16].

## 4. Discussion

### 4.1. 3D Imaging of Microtubule Filaments

Our artificial muscle specimens were constructed with microtubule filaments that included fluorescent labels, designed to facilitate imaging experiments to examine the assembly of filaments. Thereby, observation by LCMS showed a reasonable structure of the filament assembly. LCMS is good at time-lapse observation, so we are also planning to capture the contraction process of the artificial muscle, which starts contraction by the addition of ATP. This study is the first step for future studies by building a 3D structural model of a microtubule structure that will be applicable to a wider range of specimens of molecular engineered materials. In addition, the preparation of this artificial muscle mediated by self-assembly of the microtubule filaments could also be investigated by time-lapse imaging. The distribution of kinesin molecules attributed by another fluorescent dye into the network could allow direct visualization of the dynamics of the force generation process by kinesin.

Please note that the 3D volume data we used showed much larger microtubule filament diameter due to the imaging resolution as discussed in the literature [17]. It was further thickened by isosurface representation, resulting in a 500 nm diameter that could be 180 nm at the best resolution with this microscopy. Using deconvolution may allow generation of volume data; however, it also runs a risk of introducing unfavorable artifact structures. Our modeling study was also feasible for such a thickened model to trace volume connectivity, which is sometimes blurry with less intense density. After a first structure model is derived, it will be possible to refine the microtubule model to be thinner and to fit the volume density with simulated enlargement of the volume contribution.

Validation of the obtained model is a challenging study because every time a simulation is run the observed structure changes. However, once the geometrical model is constructed, the model may be fitted into the part of the new observation and possibly altered and extended by automated computer algorithms to optimize the radius and length of the cylinders. That modeling study would be used to understand and improve the physical characteristics of the bulk material, such as elasticity.

### 4.2. Undestanding the Dynamics of Microtubule Filaments

Visualizing the dynamics of the microtubule filaments required that kinesin molecules be tightly bound to the microtubule; otherwise, the filament sliding event was lost. We considered that a single kinesin multimer linker would be enough to pull the filaments, but multiple kinesin multimer linkers worked together to slide filaments. Their binding would also work as a constraint for other kinesins onto the filaments. That will require much less binding affinity, which would probably fit to the real molecular interaction of the kinesin and tubulin.

Before obtaining experimental evidence with substantial experimental cost, taking advantage of simulated motility would be useful for constructed structural models of the microtubule network. Such motility simulations of kinesin multimer linker could be calculated to evaluate the optimum affinity to yield ideal force generation conditions.

Based on this study, we propose an idea for introducing a quick recovery system to this artificial muscle. Since force generation simulation required a substantial degree of binding of kinesin molecules to the microtubules, releasing the kinesin from the microtubules could be accomplished by decreasing the binding affinity of kinesin. The rich resource of biochemical characteristics for kinesin and tubulin already includes possible chemical processing to alter the binding affinity of kinesin [18]. Although these findings were not clarified in the physiological conclusion, the functions of protein would be useful for applications in engineering of micromachines. In addition, the network structure mediated by DNA origami [5,6,19] could also be further investigated by extending this modeling study.

### 4.3. Model Building with 3D Modeling Software

Recent super resolution imaging for mammalian centriole subdistal appendages and distal appendages [20] showed a clear filament assembly structure of microtubules, with intriguing biological findings. For detailed discussions of molecular interactions on the supramolecular level, complex 3D models made by Blender were effectively used.

In this study we built a mesoscopic structure model with Blender, mainly using a polygon mesh model. Obtaining the structure of proteins and biological macromolecules on the nanoscale has been difficult experimentally, since modeling studies always lack validation. Moreover, the cost and expertise required for software tools deters laboratory researchers from undertaking 3D modeling. We presented an example of construction of a microtubule network structure including preliminary dynamics simulation using Blender. In the near future we expect the motions of such a network structure of protein including DNA will be described with atomic models, so that molecular dynamics simulations [21] would be applied to simulate the dynamics property of the molecular system. Advanced computer graphics methods that take advantage of virtual reality techniques [22] would help understand the network’s structural features. In addition, simulation studies for the bulk motion of the gel, made with the microtubule network [23,24], could also be fully understood in detail, based on our mesoscale structure model.

We have also studied a structural model of smooth muscle using Blender (Figure 6). Based on a few structural studies of smooth muscle [25] and larger knowledge of skeletal muscle [26], a plausible model of filament structure was predicted. Through the influence of a recent imaging study of cell culture of differentiated smooth muscle cells [27], the elements of the contractile unit were made using two broom-like bundles of actin filaments. The animated structure directly provides understanding of the motility mechanisms. Since this is an example of how mesoscale molecular modeling studies can substantially benefit wider molecular science studies, we also provide variations of molecular animations on our website [12].

## 5. Conclusions

Our imaging result was satisfactory for visualizing the detailed structures of the artificial muscle constructed by microtubule filaments that allowed us to construct a mesoscopic network model with possible dynamics calculation. The modeling method that we introduced effectively utilizes recent computational tools to assist construction of the mesoscale molecular model that could be shared in a research laboratory. The molecular mechanism understanding gained from such structure-based studies could play an important role in the development of artificial muscle and wider molecular science.

## Figures and Tables

**Figure 1 micromachines-11-00844-f001:**
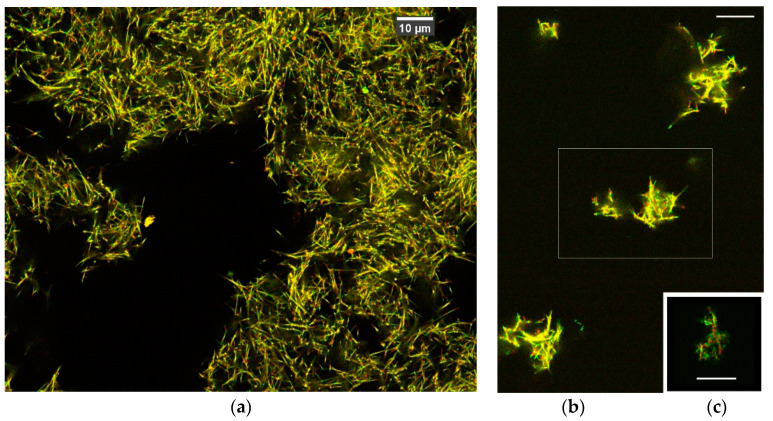
Imaging results of the network structure of microtubules in an artificial muscle specimen, by confocal laser scanning microscopy (CLSM): (**a**) the microtubule concentration at 10 μM, (**b**) the concentration at 0.1 μM. The inset box in (**b**) was used for the 3D volume data construction in this study, and (**c**) an assembly of microtubules without kinesin linkers. Microtubule filaments labeled with Alexa488 dye visible in green color, while the DNA origami binding region is labeled with TAMRA dye and is visible in red color. Most filament bodies of microtubules bind both dyes and present a yellow appearance. Scale bars, 10 μm.

**Figure 2 micromachines-11-00844-f002:**
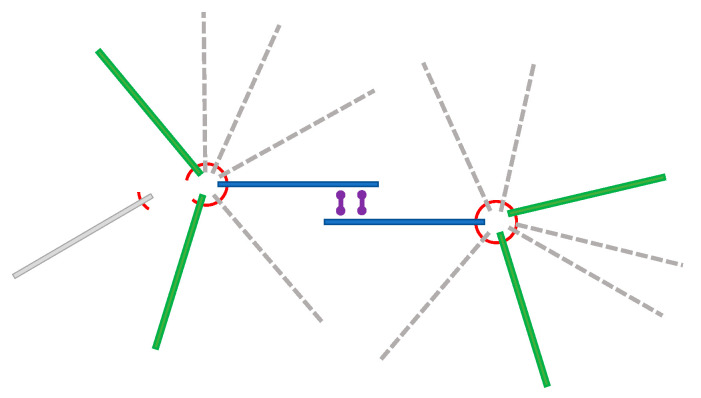
Schematic diagram of two microtubule aster structures for the contraction system. The aster structure was mediated by a DNA origami linker, indicated by red circles. The multimeric kinesin linker binds two microtubule filaments between asters to pull the asters closer, depicted as two dumbbells. Aster fragments are depicted as solid line filaments, omitting the dashed line filaments. A gray filament on the left side is an example of the dissociation that may occur to fragment the aster structure.

**Figure 3 micromachines-11-00844-f003:**
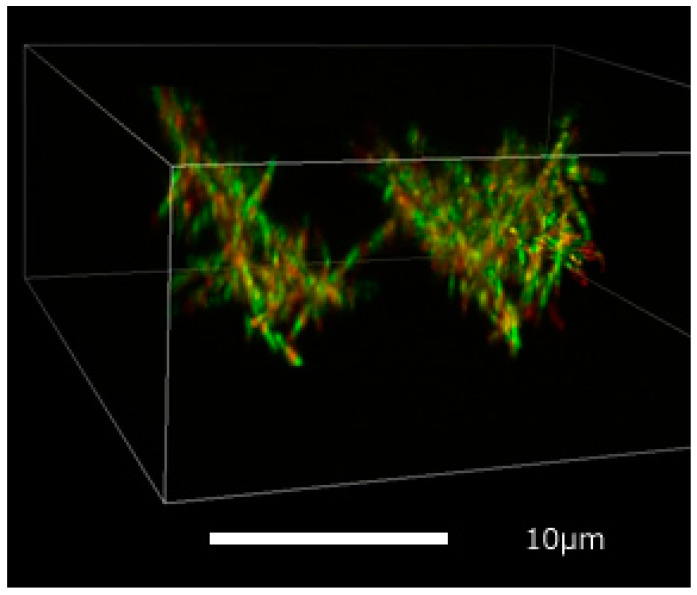
A volume rendering for the obtained 3D depth slice image dataset by CLSM. Microtubule filaments stacks were inclined toward each other. Background noise was visually decreased by low signal level cut off. The white rectangle box is 35 × 35 × 14 μm.

**Figure 4 micromachines-11-00844-f004:**
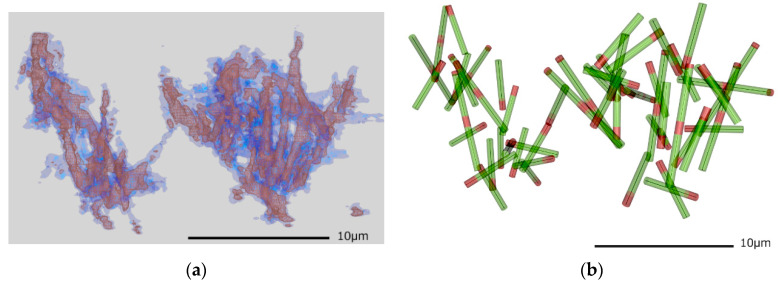
(**a**) The surface representation for obtained 3D volume data by z-slice images. Two isosurface values at high density (brown) and low density (blue) are depicted. Some portions of the blurred volumes showed less intensity. (**b**) Cylinder models of microtubules fitted to the observed volume data. We also tried to fit the direction of the cylinders, with one end marked in red, to the observation shown in Figure 3. Most network structures were well described by these cylinders, however, at regions where the cylinders intersect, it was difficult to recognize their geometrical components. The center lines for each cylinder are also depicted to indicate the real microtubule filament diameter of 25 nm. This inflation of the cylinder radius due to the resolution limit is discussed in Section 4.1.

**Figure 5 micromachines-11-00844-f005:**
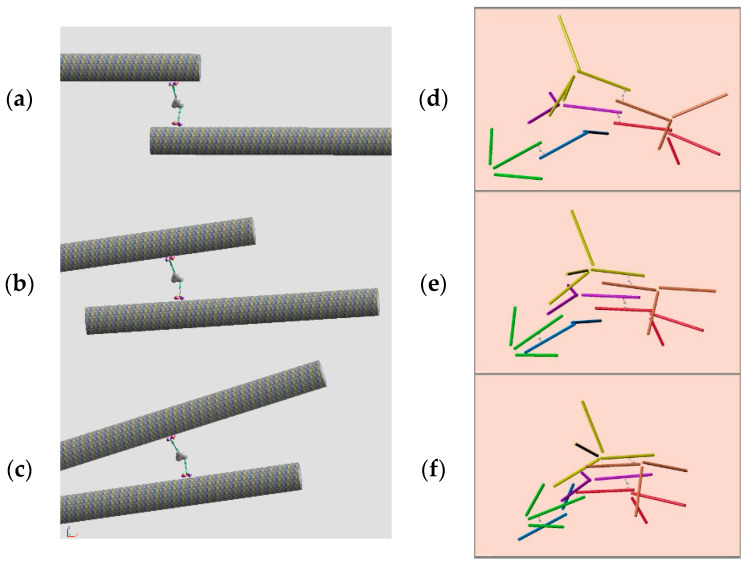
The motility element in the microtubule filaments network: the motion of multimeric kinesin linker on the two microtubule filaments at the three states of a possible contraction motion are depicted for (**a**) starting, (**b**) running, and (**c**) ending. The tentative model with three sets of fragments of microtubule asters. Applying motion to those three states of motion yields the shrinking of the network: (**d**) starting, (**e**) running, and (**f**) ending.

**Figure 6 micromachines-11-00844-f006:**
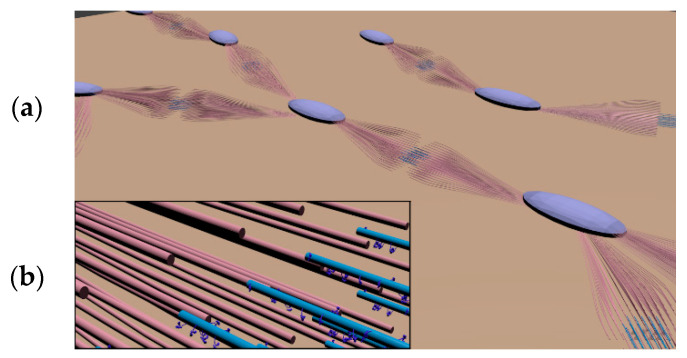
(**a**) A predicted filament structure model for smooth muscle. Cyan ellipsoids designate the dense body that binds actin filaments depicted in red. Actin filaments were bundled to form a broom-like structure. Two of these structures will hold a myosin filament with protruding myosin heads. A motor protein is depicted in blue. (**b**) A magnified view of the actin and myosin filaments interacting on the overlapped regions.

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
