# Peer review of "Modeling a Microtubule Filaments Mesh Structure from Confocal Microscopy Imaging"

_micromachines, 2020, doi:10.3390/mi11090844_

Round 1
Reviewer 1 Report
The study was presented well by the authors. The following are some minor suggestions for the authors to consider:
Line 109: im
Line 109: the previous report needs citation
Line 182: grammatical correction is needed
Line 263-266: the sentence needs grammatical correction
Line 268: meso scale should be changed to mesoscale to make it consistent throughout the paper
In addition to the minor comments above, the authors may improve the strength of their paper by discussing how their proposed model can be validated/supported by in vivo, genetic, or studies using other approaches/methodologies. The authors did mention that it was a challenge to validate their proposed model, but it may not be a bad idea to elaborate more on how this challenge can be potentially met.
Author Response
Reviewer: 1
Comments and Suggestions for Authors
The study was presented well by the authors. The following are some minor suggestions for the authors to consider:
Line 109: im
The image shown in Figure 1 <-- the typo was mended.
Line 109: the previous report needs citation
Reference [4] Matsuda et al. was added.
Line 182: grammatical correction is needed
LINE 197: Most network structures were well described by these cylinders, however, regions where cylinders intersect were difficult to recognize their geometrical components.
Line 263-266: the sentence needs grammatical correction
LINE293: In near future we expect the motions of such network structure of protein including DNA will be described with atomic models, so that molecular dynamics simulations [21] would be applied to simulate dynamics property of the molecular system. Advanced computer graphics method taking advantage of virtual reality technique [22] would help understanding its structural features.
Line 268: meso scale should be changed to mesoscale to make it consistent throughout the paper
mesoscale structure model <-- this phrase is used.
In addition to the minor comments above, the authors may improve the strength of their paper by discussing how their proposed model can be validated/supported by in vivo, genetic, or studies using other approaches/methodologies. The authors did mention that it was a challenge to validate their proposed model, but it may not be a bad idea to elaborate more on how this challenge can be potentially met.
Validation of the obtained model is a challenging study because every time observed structure changes. However once the geometrical model is constructed, the model may be fitted into the part of new observation and possibly altered and extended by automated computer algorithms optimizing radius and length of cylinders. That modeling study would be used to understanding and improving the physical characteristics of the bulk material such as elasticity. <--- This discussion is added to LINE 255.

Reviewer 2 Report
Summary
Ueno and coauthors study a microtubule and kinesin network model as potentially artificial muscle. The authors have recently published a more detailed description of the experimental setup where they observe network contraction. Here, the microtubules are partially assembled in aster fragments but do not resemble biological microtubule asters. The assembled networks were imaged by confocal microscopy and the structure and orientation of the microtubules was extracted from image stacks. Finally, the authors use a modeling approach based on kinesin dimers and fragment asters to test whether the network contraction could be reproduced in silico. Although the observed microtubule organization doesn’t recapitulate asters, the system seems interesting. The confocal imaging is not particularly high resolution but sufficient to extract the microtubule structure. Modeling such a microtubule network seems a useful approach, but it is not clear what the authors are trying to extract from the model aside form the observation that the overall networks slightly contracted. Overall this paper is an interesting start but needs more work as outlined below.
Major
It is not clear why this microtubule aster setup relates to muscle. Muscle can contract and relax in cycles while it is not clear how the system describe in this work can actually relax. If it is indeed a single contraction system, the reference to muscle is a somewhat far-fetched.
The methods section should describe the reagents and experimental procedures in much more detail. It is not clear how the different components were generated, labeled and mixed to obtain the observed meshes. It is also not clear how the 160nm resolution in the confocal images was determined. Moreover, the settings that were used for the modeling should be summarized in the text and ideally as a supplemental figure that explains the workflow. such that follow up work can reproduce the data. Lastly, the main scripts used for this work should be made available on a code sharing website.
The structures in figure 1b, at the diluted microtubule concentration do not look like organized asters. It is a random mesh of short filaments with unclear orientation. The hallmark of asters in the biological setting, is the common nucleation center for the filaments and the homogenous orientation of the filaments. This is clearly not reproduced in this model system. If they indeed observed aster like structures, the authors should provide example images. Otherwise, the reference to asters should be avoided.
The actual physics of the modeling is not described in the methods section. And the modeling results are not quantitatively assessed. The authors provide snapshots at different stages to make the case that these pseudo-asters can contract. Beside the observation that the overall mesh decreases in size, it is not clear what can be learned from this model. The authors should strive to obtain additional insight from the model and compare it to the experimental data.
The modeling results are not compared with experimental data. One would expect that the motivation for the modeling approach is this comparison. In Matsuda et al. 2019 Nano letters, the microtubule mesh shrank to about 1/3 of its diameter upon addition of ATP. They observed differences in contraction kinetics depending on presence of the crosslinking DNA origami. Would the current model reproduce these differences? The authors should discuss how consistent the modeling results are with the experimental observations. There are also some discrepancies with regards to the kinesin oligomerization state. It appears like the experimental system is based on kinesin tetramers, while the modeling employs kinesin dimers. It is not clear how that would affect the ability to compare the results from the model with the experiment.
Minor
Figure 1 and 3: the figure legend should indicate what color code was used. I seed red, green and yellow, but it is not clear what it means.
Line 109: typo in the second word. Also they reference previous work and should add the corresponding citations.
Line 161: the authors state the cylinder model “nicely” fit the data. A more quantitative approach to the comparison of model and data should be used. Figure 4 should also contain the original color information to support the orientation of the filaments.
Figure 5: Kinesin dimers are depicted in this figure. In the original Matsuda et al. 2019 Nano letters paper they used streptavidin linked kinesins that formed tetramers. It is not clear from the methods section of the text whether kinesin dimers or tetramers would be expected to be the predominant species under these experimental conditions.
First paragraph of the discussion: it needs to be rewritten. It is not clear what the authors are trying to say.
Author Response
Reviewer: 2
Major
It is not clear why this microtubule aster setup relates to muscle. Muscle can contract and relax in cycles while it is not clear how the system describe in this work can actually relax. If it is indeed a single contraction system, the reference to muscle is a somewhat far-fetched.
Currently, we successfully demonstrated dynamic contraction of microtubule network like the contraction of muscles, that is initiated by the addition of ATP as described in previous report [4]. We will also discuss potential strategy to accomplish multiple time contraction by a quick recovery method based on the obtained dynamic property of this contractile material.
<– This sentence was added to the introduction section LINE 62. As we demonstrated, there are other papers in literature that also reported artificial muscles prepared from various materials that exhibited only contraction. We agree that relaxation of the contracted network and repeated contraction-relaxation cycles like muscles are yet to demonstrate. There are strategies to accomplish multiple cycles of contraction-relaxation, and we hope that we can report some of success in near future.
The methods section should describe the reagents and experimental procedures in much more detail. It is not clear how the different components were generated, labeled and mixed to obtain the observed meshes.
The artificial muscle specimen, composed of hierarchically ordered microtubule asters, was prepared as reported previously [4]. Short microtubule seeds were polymerized from tubulin dimers labelled with fluorescent dye Alexa 488. The seed microtubules were elongated through polymerization of azide-conjugated tubulins and tubulin dimers that were labelled with TAMRA dye. Single strand DNA with a predesigned sequence was conjugated to the elongated part of the microtubules through click reaction. According to the standard procedure as reported in a literature [5] rhodamine or Alexa488-labeled tubulin was prepared using 5/6-carboxytetramethylrhodamine succinimidyl ester (TAMRA-SE;Invitrogen) or Alexa488 respectively with the standard technique and the labeling ratio was 1.0 in each case. Azide-labeled tubulin was prepared using N3-PEG4-NHS following the protocol of labeling tubulin with fluorescent dye. After preparation, the DNA-conjugated microtubules were mixed with DNA origami to form asters. Upon addition of a kinesin multimer linkers created with streptavidin molecules binding four kinesin dimers, and ATP the filament mesh structure contracted like muscle. We further examined this contraction scheme in our model calculation described below. <– More detailed descriptions for the preparation of the material was added.
It is also not clear how the 160nm resolution in the confocal images was determined. Moreover, the settings that were used for the modeling should be summarized in the text and ideally as a supplemental figure that explains the workflow. such that follow up work can reproduce the data. Lastly, the main scripts used for this work should be made available on a code sharing website.
Here we state 180 nm resolution that is listed for CLSM at the reference [9]. Although modern microscopy usually provides much better resolution acknowledged by microscopy manufacturers, for this study 180 nm resolution was sufficient. The workflow and the main scripts used for this work was already uploaded to our web site at reference [18] including obtained volume data and microtubule mesh structure model.
The structures in figure 1b, at the diluted microtubule concentration do not look like organized asters. It is a random mesh of short filaments with unclear orientation. The hallmark of asters in the biological setting, is the common nucleation center for the filaments and the homogenous orientation of the filaments. This is clearly not reproduced in this model system. If they indeed observed aster like structures, the authors should provide example images. Otherwise, the reference to asters should be avoided.
Figure 1(c) was added to show a sample aster structure of microtubule mediated DNA origami. The observation is mostly same to the previous report, Matsuda et al [4]. We consider that aster structure in both images may be fragmented by dissociation of some microtubules depicted in gray dot lines in Figure 2. The fragments of microtubules make up the mesh structure. Since the imaging result is not changed from previous report [4], we would like to describe the structure as an aster.
The actual physics of the modeling is not described in the methods section. And the modeling results are not quantitatively assessed. The authors provide snapshots at different stages to make the case that these pseudo-asters can contract. Beside the observation that the overall mesh decreases in size, it is not clear what can be learned from this model. The authors should strive to obtain additional insight from the model and compare it to the experimental data.
Using Blender, modeling geometrical objects and the physics interaction are very feasible. All the scripts were uploaded to our website and available to replay on a moderate PC.
The modeling results are not compared with experimental data. One would expect that the motivation for the modeling approach is this comparison. In Matsuda et al. 2019 Nano letters, the microtubule mesh shrank to about 1/3 of its diameter upon addition of ATP. They observed differences in contraction kinetics depending on presence of the crosslinking DNA origami. Would the current model reproduce these differences? The authors should discuss how consistent the modeling results are with the experimental observations.
We consider that Figure 5(f) showed about 2/3 shrink in diameter, however it would again shrink into 1/3 size by other kinesin multimers. The contraction model is not fully simulated, but propose the possible mechanism.
There are also some discrepancies with regards to the kinesin oligomerization state. It appears like the experimental system is based on kinesin tetramers, while the modeling employs kinesin dimers. It is not clear how that would affect the ability to compare the results from the model with the experiment.
The modeling results are used to understand how the contraction works. Because behavior of kinesin was not observed experimentally, we have to speculate the position and orientation of kinesins. It would be next study for us to compare quantitatively between observation and our theoretical model. This report focused on the method how 3D volume data are obtained and used to build the network structure of microtubule. The method using Blender was not yet widely used. So far we were not able to simulate shrinking of the microtubule network. We only demonstrate possibility for the simulation.
There are also some discrepancies with regards to the kinesin oligomerization state. It appears like the experimental system is based on kinesin tetramers, while the modeling employs kinesin dimers. It is not clear how that would affect the ability to compare the results from the model with the experiment.
Although introduced kinesin tetramer has four arms, we first calculate two arms of the kinesin heads to pulling two microtubule filaments. – The sentence was inserted at LINE 203. In fact we were not able to use four arm model. Because it is not yet known how four arms interact with other microtubules. First we only calculate a simplified case with two arms for the motility simulation as a basic motility element.
Minor
Figure 1 and 3: the figure legend should indicate what color code was used. I seed red, green and yellow, but it is not clear what it means.
Microtubules filaments labeled with Alexa488 dye visible in green color, while the DNA origami binding region is labeled with TAMRA dye and can be visible in red color. Most filament bodies of microtubules bind both dyes and present yellow appearance.<– The description was added to the Figure 1 legend.
Line 109: typo in the second word. Also they reference previous work and should add the corresponding citations.
This typo was fixed: im à image; reference [4] to previous work.
Line 161: the authors state the cylinder model “nicely” fit the data. A more quantitative approach to the comparison of model and data should be used. Figure 4 should also contain the original color information to support the orientation of the filaments.
In next study we will calculate discrepancy factor between model and data based on root means square values of obtained volume density. That will provide a “figure of merit” for the modeling tool. It also depends on the condition of the material preparation and experimental noise on imaging, and so on. We consider that obtained model simply propose an interpretation of the structure. We worked on volume display using original color information to support the orientation of the filaments, however, our data were not clear enough for the task. Currently we only used the volume isosurface of the microtubule mesh structure.
Figure 5: Kinesin dimers are depicted in this figure. In the original Matsuda et al. 2019 Nano letters paper they used streptavidin linked kinesins that formed tetramers. It is not clear from the methods section of the text whether kinesin dimers or tetramers would be expected to be the predominant species under these experimental conditions.
The kinesin multimers are designed such that each multimer will have four kinesin dimers attached to it. In our model, only two dimers are considered as shown in Figure 5. With the current model, it was not possible to use all the four dimers to interact with more than two microtubules. This is because, the positions and orientations of kinesin multimers are not confirmed experimentally, as well as the interactions of kinesin heads with microtubules. – This was also mentioned in LINE 203.
First paragraph of the discussion: it needs to be rewritten. It is not clear what the authors are trying to say.
Since our artificial muscle specimens constructed with microtubule filaments include fluorescent labels were designed to facilitate imaging experiments to examine the assembly of filaments, observation by LCMS showed a reasonable structure of the filament assembly. LCMS is good at time-lapse observation, so we are also planning to capture the contraction process of the artificial muscle, which starts contraction by the addition of ATP. This study is the first step for these future studies … <-- The beginning sentences LINE 235 are rewriting.

Reviewer 3 Report
The authors applied two-color laser scanning confocal microscopy system to visualize microtubules. Acquired 3D data was processed with the general‐purpose 3D modeling software Blender. This is an interesting study. Manuscript is well written and provides detailed information so that readers can follow the steps.
However, visualizing microtubule with “real” super-resolution methods was already published. (Super-resolution microscopy reveals coupling between mammalian centriole subdistal appendages and distal appendages, eLife 2020) Data was also processed by Blender. What is missing in this manuscript is to clarify the meaning of the work. Otherwise, it will look like a repeated study with inferior technique.
The term “super-resolution” can be misleading as readers may expect methods such as STORM, PALM or STED. Deconvolution microscopy, although it can marginally improve imaging quality, is not considered as a super-resolution technique. Careful proof reading is required. (2.4 is missing, im => image in Result section)
Author Response
Reviewer: 3
Comments and Suggestions for Authors
The authors applied two-color laser scanning confocal microscopy system to visualize microtubules. Acquired 3D data was processed with the general‐purpose 3D modeling software Blender. This is an interesting study. Manuscript is well written and provides detailed information so that readers can follow the steps.
However, visualizing microtubule with “real” super-resolution methods was already published. (Super-resolution microscopy reveals coupling between mammalian centriole subdistal appendages and distal appendages, eLife 2020) Data was also processed by Blender. What is missing in this manuscript is to clarify the meaning of the work. Otherwise, it will look like a repeated study with inferior technique.
We add the paper to reference [18] and discuss about the difference at LINE 283. Their study is a good example of recent super-resolution microscopy to explore biological mystery. They illustrated the super molecular complex structure by CG model with Blender. That is exactly what we recommend Blender for scientific study. In this study we are demonstrating advanced use of Blender, i.e. volumetric data analysis and rigid body physics calculation to understand obtained experimental data.
The term “super-resolution” can be misleading as readers may expect methods such as STORM, PALM or STED. Deconvolution microscopy, although it can marginally improve imaging quality, is not considered as a super-resolution technique. Careful proof reading is required. (2.4 is missing, im => image in Result section)
That word in LINE 39 and 59 was changed to “high-resolution”. I agree that our method may not always be regarded as a super-resolution. The section number 2.4 and typo, im, was mended.

Round 2
Reviewer 2 Report
Summary
The authors addressed some of my concerns. However, the manuscript still needs additional major revisions as outlined below.
Major
The authors claim that the modeling with the Blender software is accessible. But if the they think that the academic contribution of this paper is the actual modeling as they are stating in the title:
Modeling a microtubule filaments mesh structure from confocal microscopy imaging
The first sentence of the abstract also suggest that the main contribution here is the modeling:
This study introduces a modeling method for a supermolecular structure of microtubules for the development of a force generation material using motor proteins.
But in my understanding the model is not described detailed enough in this paper. E.g. there must be model parameters such as kinesin motor speed and length scales of the objects and they should all be provided in a table. Next, the physics behind the model has to be given as equations in the methods section. Also, given a certain motor speed it is not clear how long the simulation in figure 5 was run. The authors should indicate the time scale. Moreover, the authors then say that the code of the model can be downloaded from their website. But we all know that such website contents changes and get rapidly lost. In order to make this work an academic contribution, the workflow of the code has to be described e.g. as supplement workflow figure and/or the code should be provided as supplement files or deposited at e.g. github. The authors cannot claim to present a modeling approach and actually not describe that modeling approach in detail and simply refer to a couple files downloadable from their website.
The authors respond that the orientation of the microtubules in figure 4 was not obtained from the image data but randomly assigned. They should add this information in the methods section. The legend in line 196 is somewhat misleading. It should clearly state that the orientation was randomly assigned.
The methods section still lacks some detail for readers who have no access to their previous Nano Letters publication.
Minor
Figure 1b and c: the length of the scale bar is not indicated.
Figure 3a: needs a scale bar and a color code should be indicated in the legend.
Figure 4a: the scale bar is missing
There are plenty of English language issues that need to be improved.
Round 3
Reviewer 2 Report
Panel 4B: the microtubule thickness is 25nm. In the current panel, the thickness appears to be almost 1 micron. The filament thickness should be drawn to the appropriate scale.
The github link should be presented in the manuscript
Author Response
Author's Reply to the Review Report (Reviewer 2)
Figure 4 (b) was modified to indicate that the real microtubule filament size is almost a thin line in the picture. The legend was also added as follows:
The center lines for each cylinders are also depicted to indicate the real microtubule filament diameter of 25 nm. This inflation of the cylinder radius due to resolution limit is discussed in section 4.1.
Our reference [16] is directed to the http://github.com/uenoyt/abam/
We also would like to add a description of our previous Nano Letters paper that should be included in this manuscript.
This material was made possible by using DNA origami technology [5,6] to bind microtubule filaments to forms the aster structure. <--- We added this sentence at LINE 34 in Introduction to clearly indicate that red portions of Figure 4 shows the DNA origami bindings as described at LINE 134.